# Beyond Supervised Learning: Recognizing Unseen Attribute-Object Pairs with Vision-Language Fusion and Attractor Networks

## Abstract

This paper handles a challenging problem, unseen attribute-object pair recognition, which asks a model to simultaneously recognize the attribute type and the object type of a given image while this attribute-object pair is not included in the training set. In the past years, the conventional classifier-based methods, which recognize unseen attribute-object pairs by composing separately-trained attribute classifiers and object classifiers, are strongly frustrated. Different from conventional methods, we propose a generative model with a visual pathway and a linguistic pathway. In each pathway, the attractor network is involved to learn the intrinsic feature representation to explore the inner relationship between the attribute and the object. With the learned features in both pathways, the unseen attribute-object pair is recognized by finding out the pair whose linguistic feature closely matches the visual feature of the given image. On two public datasets, our model achieves impressive experiment results, notably outperforming the state-of-the-art methods.

## 1 Introduction

Inferring the unknown from the known signals the advanced intelligence. This paper targets to teach the machine to recognize unseen attribute-object pairs based on seen attribute-object pairs. For example, teach the machine to recognize the "sliced apple" (unseen pairs) after letting the machine observe the samples of "green apple" and "sliced tomato" (seen pairs), as shown in Fig. 1 (upper). This problem is termed as unseen attribute-object pair recognition.

Unseen attribute-object pair recognition is a meaningful and challenging problem. Deep neural network techniques (Krizhevsky et al., 2012; Lecun et al., 2015; Simonyan & Zisserman, 2015; He et al., 2015), especially supervised learning techniques, have achieved impressive successes on various tasks. However, due to the large number of possible attribute-object pairs, supervised learning methods, asking for massive data annotations for each attribute-object pair, will confront the composition explosion disaster. Therefore, it is meaningful to recognize unseen attribute-object pairs based on seen pairs. The main challenge of the unseen attribute-object pair recognition is that the testing attribute-object pairs are not included in the training set. The abstractness of attributes further increases the challenge. Another challenge results from the similarity between some attributes such as "huge" and "big".

To recognize unseen attribute-object pairs, conventional methods (Chen & Grauman, 2014; Misra et al., 2017) typically learn attribute classifiers and object classifiers at the first, and then recognize unseen pairs by composing these separately-trained classifiers, which ignore the inner relationship between attributes and objects. The conventional classifier-based methods also ignore the abstractness of attributes. It is the fact that objects can be accurately classified since the same type of objects share similar appearance. As shown in Fig. 1 (middle), different cars present similar characteristic. However, the attribute is abstract, and the same type of attribute varies significantly when describing different types of objects. For example, as shown in Fig. 1 (lower), the visual characteristics of the attribute "beautiful" are notably different from each other when describing a sunset, an aurora, or a mountain. Due to the abstractness of attributes, the classifier-based methods achieve lower performance on attribute recognition than that on the object recognition, which further leads to the low

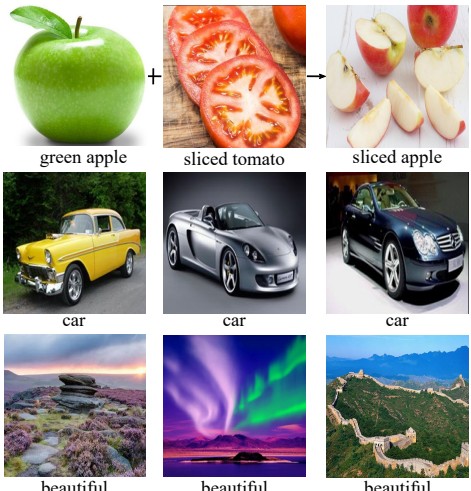

Figure 1: Problem definition and analysis.

accuracy of attribute-object pair recognition. In addition, though an image conveys rich information in internal structure, the conventional classifier-based methods are designed to learn a feature representation to simply infer a label, and any information not required to infer the label is omitted. Unseen attribute-object pair recognition is a high-level and complex vision problem relying on the deep and comprehensive understanding of an image, which asks a model to learn an intrinsic feature representation conveying the rich information of an image.

Based on the above observations, we propose a dual pathway generative model. The dual pathways are the visual pathway and the linguistic pathway, respectively. In each pathway, taking the initial feature representation as the input, we employ the attractor networks to iteratively learn the intrinsic feature representation. We design several loss functions to optimize the model. Given the optimized model and all possible attribute-object pairs, the recognition is realized using a voting inference method. In the experiments, we compare our model with several state-of-the-art models on two challenging public datasets. Experiment results demonstrate that our model outperforms the state-of-the-art models by large margins.

The contributions of this paper are as follows: (1) We for the first time introduce the attractor networks to recognize unseen attribute-object pairs; (2) Our method exhibits much better performance than conventional methods and state-of-the-art methods on two challenging public datasets.

## 2 RELATED WORK

**Unseen Attribute-Object Pair Recognition** The intuitive idea to recognize unseen attribute-object pairs is combining attribute classifiers and object classifiers (Chen & Grauman, 2014; Misra et al., 2017). However, these classifier-based methods separately process the attribute and the object, ignoring the inner relation between the attribute and the object. As a result, these methods do not achieve satisfied performance and heavily suffer from the "domain shift" problem (Fu et al., 2015) - the distribution of the testing data is different from that of the training data. To overcome the "domain shift" problem, Nagarajan & Grauman (2018) creatively proposes to model an attribute as an *operator* and an attribute-object pair as an object vector that is "operated" by the *operator*, and the model presents competitive results. Recently, Nan et al. (2019) introduces a generative model with the encoder-decoder mechanism and obtains state-of-the-art results. Motivated by these works, we propose a generative model with attractor networks and vision-language fusion mechanism to recognize unseen attribute-object pairs.

**Semantic Attractor Network** The attractor network is a recurrent neural network, which is originally proposed as a memory model to represent a concept (Hinton & Shallice, 1991; Herrmann et al., 1993). The model implies that a concept in the semantic memory is represented by a set of nodes that are mutually connected according to their semantic relatedness (Lerner et al., 2010). Given the

initial nodes to represent a concept, the value of each node evolves over time to reach a stable state, which is called an attractor (Plaut & Shallice, 1993; Masson, 1995; Cree et al., 1999). A set of states surrounding an attractor is termed as the basin of attraction (Zemel & Mozer, 2001; ichi Asakawa, 2013). The basin of attraction plays a key role in the evolution of an attractor network, and the states surrounding an attractor represent similar concepts.

For unseen attribute-object pair recognition, a model is required to learn an intrinsic representation of the given image. The attractor network is able to recurrently learn a set of mutually connected nodes to represent the input image, and this representation is stable and intrinsic. In addition, the basin of attraction allows the model to associate similar unseen attribute-object pairs with the seen pairs, which is significant for unseen attribute-object pair recognition. Therefore, we involve attractor networks in our model to further improve the stability and generalization of the initial feature representations in both visual pathway and linguistic pathway.

**Zero Shot Learning** The supervised learning is one of the most important techniques in artificial intelligence, presenting striking advantages on a series of tasks in object detection (Sermanet et al., 2013), machine translation (Bahdanau et al., 2014) and speech recognition (Graves et al., 2013). However, the supervised learning heavily relies on massive data annotations. The large scale data annotation is time-consuming, and some data are difficult to annotate. Therefore, zero shot learning (ZSL) gradually draws researchers' interest (Larochelle et al., 2008; Palatucci et al., 2009). ZSL aims to recognize unseen objects (i.e., object types in the testing set are not included in the training set). To recognize unseen objects, early works including Lampert et al. (2009; 2014) typically learn a projection from the input visual space to a semantic space where the attribute descriptions of unseen objects are known. Then, given an image, the model extracts visual feature in the visual space and projects it into the semantic space to obtain the attribute prediction. The recognition is realized by finding the unseen object whose attribute description is closest with the attribute prediction. For these methods, attribute classifiers are trained separately and the relation between attributes are ignored. In order to mitigate this issue, some works seek to embed the visual features and the attribute descriptions into a common latent space (Ba et al., 2015; Wang et al., 2019). However, the input visual space and the latent space have their own manifold structure, which usually leads to significant variation in recognition performance. Therefore, Li et al. (2017) proposes an alternate optimization mechanism to make the latent space consistent with the input space. However, the above methods lack either the ability to learn the bi-directional mappings between the visual space and the semantic/latent space or a flexible metric to evaluate the similarity between different kinds of features, thus Huang et al. (2018) leverages the Generative Adversarial Network (GAN) to generate various visual features conditioned on class labels and maps each visual feature to its corresponding semantic feature, then recognizes unseen objects by measuring the similarity between the visual feature and textual feature.

## 3 APPROACH

### 3.1 PROBLEM FORMULATION

For unseen attribute-object pair recognition, the training set (seen pairs) is defined as $\mathcal{S} = \{x_i^s, (a_i^s, o_i^s)\}_{i=1}^{n_s}$, where $x_i^s$ is the $i^{th}$ image, and $(a_i^s, o_i^s) \in \mathcal{L}_s$ is its corresponding attribute label and object label. The testing set (unseen pairs) is defined as $\mathcal{U} = \{x_j^u, (a_j^u, o_j^u)\}_{j=1}^{n_u}$, where $x_j^u$ is the $j^{th}$ image, and $(a_j^u, o_j^u) \in \mathcal{L}_u$ is its corresponding attribute label and object label. The seen pairs and unseen pairs are disjoint, i.e., $\mathcal{L}_s \cap \mathcal{L}_u = \emptyset$. During the testing, we identify the attribute-object pair label of a given image from all possible pairs $\{(a_j^u, o_j^u)\}_{j=1}^{n_u}$, where $n_u$ is the number of all possible pairs.

### 3.2 NETWORK ARCHITECTURE AND DATA TRANSITIONS

The overview of our network architecture is illustrated in Fig. 2. The model consists of the visual pathway to process the visual data and the linguistic pathway to process the linguistic data. For the visual pathway, given an input image, a Convolution Neural Network (CNN) is used to extract the initial visual feature $x^\mathcal{V}$, which is further processed by the visual encoding module, obtaining the visual encoder feature $e^\mathcal{V}$. $e^\mathcal{V}$ serves as the input of an attractor network, which outputs the

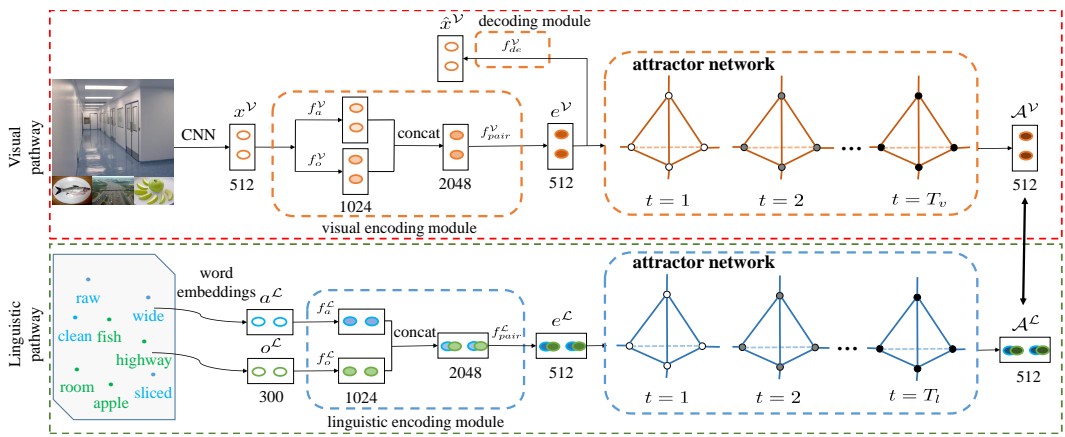

Figure 2: The overview of our network architecture. The network is composed of the visual pathway and the linguistic pathway. In the visual pathway, an input image is processed by the Convolution Neural Network (CNN), the visual encoding module and an attractor network, obtaining the visual attractor feature $\mathcal{A}^{\mathcal{V}}$. In the linguistic pathway, the data flows are different for the training procedure and testing procedure. During the training, the linguistic attribute feature $a^{\mathcal{L}}$ and the linguistic object feature $o^{\mathcal{L}}$ that correspond to the input image are firstly extracted, and then processed by the linguistic encoding module and an attractor network, obtaining the linguistic attractor feature $\mathcal{A}^{\mathcal{L}}$. The loss measuring the similarity between $\mathcal{A}^{\mathcal{V}}$ and $\mathcal{A}^{\mathcal{L}}$ as well as other losses are computed to train the neural network. During the testing, the linguistic attractor features of all possible unseen attribute-object pairs are computed (for simplicity, only one pair is illustrated in the figure) and compared with the visual attractor feature of the input image to predict the most likely attribute-object pair.

visual attractor feature $\mathcal{A}^{\mathcal{V}}$. $e^{\mathcal{V}}$ also serves as the input of the decoding module, which outputs the reconstruction visual feature $\hat{x}^{\mathcal{V}}$.

For the linguistic pathway, the attribute word and the object word are embedded as the linguistic attribute feature $a^{\mathcal{L}}$ and the linguistic object feature $o^{\mathcal{L}}$, respectively. $a^{\mathcal{L}}$ and $o^{\mathcal{L}}$ are processed by the linguistic encoding module, obtaining the linguistic encoder feature $e^{\mathcal{L}}$. $e^{\mathcal{L}}$ serves as the input of an attractor network, which outputs the linguistic attractor feature $\mathcal{A}^{\mathcal{L}}$.

**Visual Encoding Module**    Given the initial visual feature $x^{\mathcal{V}}$, the visual encoder feature $e^{\mathcal{V}}$ is computed as:

$$e^{\mathcal{V}} = f_{pair}^{\mathcal{V}}(concat[f_a^{\mathcal{V}}(x^{\mathcal{V}}), f_o^{\mathcal{V}}(x^{\mathcal{V}})]) \tag{1}$$

where $concat$ represents the concatenation operation, $f_a^{\mathcal{V}}(\cdot)$, $f_o^{\mathcal{V}}(\cdot)$, and $f_{pair}^{\mathcal{V}}(\cdot)$ are three linear functions.

**Visual Attractor Network**    As mentioned above, an attractor network consists of a set of mutually-connected nodes. Given the initial value of all nodes, the attractor network evolves over time to update the value of each node and finally reaches a stable state. We use the visual encoder feature $e^{\mathcal{V}}$ to initialize all nodes, and follow the same update equations described in the work of Devereux et al. (2018) to iteratively update their values.

In detail, for a node $n$, let $q_n(t)$ be the input of the node $n$ at time $t$ and $a_n(t)$ be the activation value of the node $n$ at time $t$. $a_n(t)$ is computed by a sigmoid function $\sigma(\cdot)$ which takes $q_n(t)$ as the input: $a_n(t) = \sigma(q_n(t))$. $q_n(t)$ is composed of two parts, the inner input and the external input. The inner input is the input of the node $n$ at time $t-1$, denoted as $q_n(t-1)$. The external input at time $t$ is a weighted sum of the activation values of other nodes connected to the node $n$, denoted as $p_n(t)$.

The external input $p_n^{\mathcal{V}}(t)$ of the node $n$ at time $t$ in the visual pathway is defined as:

$$p_n^{\mathcal{V}}(t) = \sum_{i \neq n} w_{in} a_i^{\mathcal{V}}(t-1) + b_n \tag{2}$$

where $w_{in}$ represents the connection weight from node $i$ to node $n$, $a_i^{\mathcal{V}}(t-1)$ represents activation value of node $i$ at time $t-1$ in the visual pathway, and $b_n$ represents the bias of node $n$.

The input $q_n^{\mathcal{V}}(t)$ of the node $n$ at time $t$ in the visual pathway is calculated as a linear combination:

$$q_n^{\mathcal{V}}(t) = c_v p_n^{\mathcal{V}}(t) + (1 - c_v)q_n^{\mathcal{V}}(t-1) \tag{3}$$

where $q_n^{\mathcal{V}}(t-1)$ represents the input of the node $n$ at time $t-1$ in the visual pathway, and $c_v$ is a proportion parameter.

The activation value $a_n^{\mathcal{V}}(t)$ of the node $n$ at time $t$ in the visual pathway is calculated as a sigmoid function $\sigma(\cdot)$ of its input $q_n^{\mathcal{V}}(t)$:

$$a_n^{\mathcal{V}}(t) = \sigma(q_n^{\mathcal{V}}(t)) \tag{4}$$

Using $\mathcal{Q}^{\mathcal{V}}(0)$ and $\mathcal{A}^{\mathcal{V}}(0)$ to represent the initial input and the initial activation value of all nodes of the attractor network in the visual pathway, we initialize the attractor network's states by setting $\mathcal{Q}^{\mathcal{V}}(0) = e^{\mathcal{V}}, \mathcal{A}^{\mathcal{V}}(0) = \sigma(e^{\mathcal{V}})$. Once initialized, the attractor network iterates recurrently according to the update equations defined in Eq. 2, Eq.3 and Eq.4 to the time-tick $T_v$, and outputs the final activation value, which is denoted as $\mathcal{A}^{\mathcal{V}}$.

**Visual Decoding Module**     For the decoding module, we reconstruct the original visual feature $x^{\mathcal{V}}$ by a linear function:

$$\hat{x}^{\mathcal{V}} = f_{de}^{\mathcal{V}}(e^{\mathcal{V}}) \tag{5}$$

**Linguistic Encoding Module**     Given the linguistic attribute feature $a^{\mathcal{L}}$ and the linguistic object feature $o^{\mathcal{L}}$, the linguistic encoder feature $e^{\mathcal{L}}$ is computed as:

$$e^{\mathcal{L}} = f_{pair}^{\mathcal{L}}(concat[f_a^{\mathcal{L}}(a^{\mathcal{L}}), f_o^{\mathcal{L}}(o^{\mathcal{L}})]) \tag{6}$$

where $concat$ represents the concatenation operation, $f_a^{\mathcal{L}}(\cdot)$, $f_o^{\mathcal{L}}(\cdot)$, and $f_{pair}^{\mathcal{L}}(\cdot)$ are three linear functions.

**Linguistic Attractor Network**     Similar with the visual attractor network, we initialize the linguistic attractor network by setting $\mathcal{Q}^{\mathcal{L}}(0) = e^{\mathcal{L}}, \mathcal{A}^{\mathcal{L}}(0) = \sigma(e^{\mathcal{L}})$, where $\mathcal{Q}^{\mathcal{L}}(0), \mathcal{A}^{\mathcal{L}}(0)$ represent the initial input and the initial activation value of all nodes of the linguistic attractor network. For a node $n$, its external input $p_n^{\mathcal{L}}(t)$ at time $t$ is a weighted sum:

$$p_n^{\mathcal{L}}(t) = \sum_{i \neq n} w_{in} a_i^{\mathcal{L}}(t-1) + b_n \tag{7}$$

where $w_{in}$ represents the connect weight from node $i$ to node $n$, $a_i^{\mathcal{L}}(t-1)$ represents the activation value of node $i$ at time $t-1$, and $b_n$ represents the bias of node $n$.

The input $q_n^{\mathcal{L}}(t)$ of the node $n$ at time $t$ is calculated as:

$$q_n^{\mathcal{L}}(t) = c_l p_n^{\mathcal{L}}(t) + (1 - c_l)q_n^{\mathcal{L}}(t-1) \tag{8}$$

where $q_n^{\mathcal{L}}(t-1)$ represents the input of node $n$ at time $t-1$ and $c_l$ is a proportion parameter.

The activation value $a_n^{\mathcal{L}}(t)$ of the node $n$ at time $t$ is calculated as:

$$a_n^{\mathcal{L}}(t) = \sigma(q_n^{\mathcal{L}}(t) \tag{9}$$

Given the initial values of all nodes, the attractor network recurrently iterates according to update equations defined in Eq. 7, Eq.8 and Eq.9 to the time-tick $T_l$, and outputs the final activation value, which is denoted as $\mathcal{A}^{\mathcal{L}}$.

### 3.3    LOSS FUNCTIONS

**Attractor Loss**     Let $\mathcal{W}$ be the matrix that represents the connection weights of all nodes in an attractor network and $w_{ij}$ be the connection weight from the node $i$ to the node $j$. The connection weight from the node $i$ to the node $j$ is required to be the same with the weight from $j$ to $i$ (i.e.,

$w_{ij} = w_{ji}$). In addition, any node in an attractor network does not connect with itself (i.e., $w_{ii} = 0$). Therefore, the attractor loss is defined as:

$$L_{at} = ||\mathcal{W} - \mathcal{W}^T||_2 + ||diag(\mathcal{W})||_2 \tag{10}$$

where $diag(\cdot)$ is a function to extract the diagonal elements in the matrix $\mathcal{W}$, and $|| \cdot ||_2$ is the L2 norm.

**Conditional Loss**  The attractor network recurrently iterates over time, and small differences in initial conditions may yield notable fluctuation of the output. Hence, we define the conditional loss $L_{cd}$ to constrain $e^{\mathcal{V}}$ and $e^{\mathcal{L}}$ to be close to each other:

$$L_{cd} = ||e^{\mathcal{V}} - e^{\mathcal{L}}||_2 \tag{11}$$

**Fusion Loss**  To fuse the information in the visual pathway and linguistic pathway, we design the fusion loss to minimize the L2 distance between the visual attractor feature $\mathcal{A}^{\mathcal{V}}$ and the linguistic attractor feature $\mathcal{A}^{\mathcal{L}}$:

$$L_{fus} = ||\mathcal{A}^{\mathcal{V}} - \mathcal{A}^{\mathcal{L}}||_2 \tag{12}$$

**Decoding Loss**  Inspired by recent works for unseen attribute-object pair recognition (Nan et al., 2019) and zero-shot learning (Kodirov et al., 2017), we introduce the decoding loss to minimize the L2 distance between the initial visual feature $x^{\mathcal{V}}$ and the reconstruction visual feature $\hat{x}^{\mathcal{V}}$ defined in Eq. 5:

$$L_{de} = ||x^{\mathcal{V}} - \hat{x}^{\mathcal{V}}||_2 \tag{13}$$

The motivation of designing the decoding loss is to learn better visual encoder feature $e^{\mathcal{V}}$.

**Discriminative Loss**  For some pairs, either the attribute or the object feature is dominant to represent the whole pair. Therefore, to preserve the individual property of the attribute and object, we design the discriminative loss, which is defined as a cross entropy loss:

$$L_{dis} = -Y^a \log(y_a^{\mathcal{V}}) - Y^o \log(y_o^{\mathcal{V}}) \tag{14}$$

where $Y^a$ denotes the attribute ground truth, $y_a^{\mathcal{V}}$ denotes the attribute prediction, $Y^o$ denotes the object ground truth, and $y_o^{\mathcal{V}}$ denotes the object prediction. $y_a^{\mathcal{V}}$ and $y_o^{\mathcal{V}}$ are computed as:

$$y_a^{\mathcal{V}} = f_{s\_a}^{\mathcal{V}}(f_a^{\mathcal{V}}(x^{\mathcal{V}})) \tag{15}$$

$$y_o^{\mathcal{V}} = f_{s\_o}^{\mathcal{V}}(f_o^{\mathcal{V}}(x^{\mathcal{V}})) \tag{16}$$

where $f_{s\_a}^{\mathcal{V}}(\cdot)$ and $f_{s\_o}^{\mathcal{V}}(\cdot)$ are two softmax functions, and $f_a^{\mathcal{V}}(\cdot)$ and $f_o^{\mathcal{V}}(\cdot)$ are two linear functions.

### 3.4 Learning and Inference

Let $W$ be all parameters of our proposed model. During the training, we learn the parameters by minimizing the loss functions:

$$W^* = \arg\min_W \alpha L_{at} + \eta L_{fus} + \gamma L_{de} + \lambda L_{dis} + \beta L_{cd} \tag{17}$$

We use the ADAM algorithm (Kingma & Ba, 2014) to learn the parameters.

During the inference, we propose a voting inference method. For an input image, its visual attractor feature $\mathcal{A}^{\mathcal{V}}$ is computed, at the same time, the linguistic attractor features of all $n_u$ possible unseen attribute-object pairs are computed, which are denoted as $\{\mathcal{A}_j^{\mathcal{L}}\}_{j=1}^{n_u}$. By computing the L2 distance between $\mathcal{A}^{\mathcal{V}}$ and each $\mathcal{A}_j^{\mathcal{L}}$, the pair that corresponds to the minimum distance is taken as the initial attribute-object pair recognition for the input image. For the images belonging to the same pair, we use the recognitions for all images to vote a pair label to be the final attribute-object pair recognition.

## 4 Experiments

### 4.1 Datasets

We evaluate the model on two public challenging datasets, the MIT-States dataset (Isola et al., 2015) and the UT-Zappos50K dataset (Yu & Grauman, 2014). The MIT-States dataset is composed of

63,440 images, covering 115 attribute classes, 245 object classes, and 1,962 attribute-object pairs. Each image is annotated with an attribute-object pair label like "dirty kitchen". Same with the setting in previous works (Misra et al., 2017; Nagarajan & Grauman, 2018; Nan et al., 2019), 1262 pairs are used for the training and 700 pairs for the testing. UT-Zappos50K is a fine-grained shoes dataset including 16 attribute classes and 12 object classes, totally with 50,025 images. Same with the setting in previous works (Misra et al., 2017; Nagarajan & Grauman, 2018; Nan et al., 2019), 83 pairs are used for the training and 33 pairs for the testing.

## 4.2 BASELINES AND METRIC

Five baseline methods, including two conventional methods and three state-of-the-art methods, are compared with our method. We briefly introduce the baselines as follows:

—ANALOG (Chen & Grauman, 2014) is a conventional method that recognizes unseen attribute-object pairs using a set of seen object-specific attribute classifiers;

—REDWINE (Misra et al., 2017) is a conventional method that recognizes unseen attribute-object pairs by composing attribute and object classifiers;

—SAE (Kodirov et al., 2017) is a state-of-the-art ZSL method. It firstly projects the input feature into a semantic space where the auxiliary information of unseen pairs is known, and the recognition is realized by finding the pair whose auxiliary information is closest with the input feature;

—ATTOPERATOR (Nagarajan & Grauman, 2018) is a state-of-the-art method that predicts unseen pairs by comparing the visual feature of the given image with all possible attribute-object features that are modeled as the object vectors transformed by attribute *operators*;

—GENERATE (Nan et al., 2019) is a state-of-the-art method that predicts unseen pairs by comparing the visual feature of the given image with the linguistic features of all possible pairs in a latent space.

We use the top-1 accuracy as evaluation metric, which is widely adopted by the state-of-the-art methods (Nagarajan & Grauman, 2018; Nan et al., 2019).

## 4.3 IMPLEMENTATION DETAILS

We use the ResNet-18 (He et al., 2015) network pretrained on the ImageNet dataset (Deng et al., 2009) to extract the 512-dimension visual feature $x^{\mathcal{V}}$. For fair comparison, neither the fine-tuning operation nor data augmentation is applied to our method and baseline methods. We use the GloVe model (Pennington et al., 2014) to extract the 300-dimension linguistic attribute feature $a^{\mathcal{L}}$ and linguistic object feature $o^{\mathcal{L}}$. In the visual encoding module and linguistic encoding module, the attribute feature and object feature are enlarged as 1024-dimension features to improve the representation potentiality, which are then concatenated as 2048-dimension features to represent attribute-object pairs. To relieve the computation burden of attractor networks, $e^{\mathcal{V}}$ and $e^{\mathcal{L}}$ are transformed as 512-dimension features to serve as the input of attractor networks, which output the 512-dimension attractor features $\mathcal{A}^{\mathcal{V}}$ and $\mathcal{A}^{\mathcal{L}}$.

Our model is implemented using TensorFlow (Abadi et al., 2016). Every linear function involved in Eq. 1, Eq. 5, Eq. 6, and Eq. 16 is implemented by one fully connected layer. During the training, the initial learning rate is 0.002, which decays by 0.95 every epoch. The batch size is 128 using a Nvidia 2080 GPU. Using the grid-search method, we set parameters $\alpha, \eta, \gamma, \lambda, \beta$ in Eq.17 to be $1.0, 2.0, 6.0, 3.0, 2.0$ for the MIT-States dataset and $1.0, 3.0, 6.0, 1.0, 2.0$ for the UT-Zappos50K dataset. Attractor parameters $c_v, c_l, T_v, T_l$ are set as $0.1, 0.2, 20, 10$ for the MIT-States dataset and $0.08, 0.16, 25, 15$ for the UT-Zappos50K dataset. The training is terminated (80 epochs on the MIT-States dataset and 120 epochs on the UT-Zappos50K dataset) when the sum of losses slightly decreases. The weight connections and biases of all nodes in an attractor are randomly initialized.

## 4.4 EXPERIMENT DESIGN AND RESULT ANALYSIS

**Compare with Baselines** The comparative results with baselines are summarized in Tab. 1. We can observe that our method outperforms the baseline methods by large margins, achieving $114.6\%$

Table 1: Top-1 unseen attribute-object recognition accuracies of baseline methods and our method on the MIT-States dataset and UT-Zappos50k dataset.

| Methods | MIT-States(%) | UT-Zappos(%) |
|---------|---------------|--------------|
| CHANCE | 0.14 | 3.0 |
| ANALOG(Chen & Grauman, 2014) | 1.4 | 18.3 |
| SAE (Kodirov et al., 2017) | 14.0 | 31.0 |
| REDWINE (Misra et al., 2017) | 12.5 | 40.3 |
| OPERATOR (Nagarajan & Grauman, 2018) | 14.2 | 46.2 |
| GENERATE (Nan et al., 2019) | 17.8 | 48.3 |
| our | **38.2** | **73.7** |

Table 2: Top-1 unseen attribute-object recognition accuracies of the "Base" model and "Base + Attractor" model on the MIT-States dataset and UT-Zappos50K dataset. "Base" denotes the model without attractor networks, and "Base + Attractor" denotes the model with attractor networks.

| Models | MIT-States(%) | UT-Zappos(%) |
|--------|---------------|--------------|
| Base | 36.1 | 51.8 |
| Base + Attractor (our) | **38.2** | **73.7** |

Table 3: Top-1 accuracies of different loss function compositions on the MIT-States dataset and UT-Zappos50k dataset. "at", "fus", "de", "dis", and "cd" represent the attractor loss $L_{at}$, fusion loss $L_{fus}$, decoding loss $L_{de}$, discriminative loss $L_{dis}$, and conditional loss $L_{cd}$, respectively.

| Losses | MIT-States(%) | UT-Zappos(%) |
|--------|---------------|--------------|
| at+fus (basic loss) | 0.14 | 7.8 |
| +dis | 0.20 | 11.2 |
| +cd | 0.21 | 12.3 |
| +de | 35.0 | 68.0 |
| +dis+cd | 0.3 | 15.1 |
| +dis+de | 36.4 | 68.9 |
| +cd+de | 36.8 | 72.4 |
| +dis+cd+de (our) | **38.2** | **73.7** |

and 52.6% accuracy improvements over the second best method on the MIT-States dataset and on the UT-Zappos50K dataset. The reasons are two-fold: 1) the attractor networks recurrently learn the intrinsic and stable feature representations, allowing the deep understanding of an image; 2) the encoder-decoder mechanism improves the robustness of feature representations, which is significant for alleviating the "domain shift" problem.

**Attractor Significance**    This experiment is targeted to validate the effectiveness of attractor networks by comparing the recognition accuracies of the base model (without attractor networks) and our model (with attractor networks). The results are shown in Tab. 2, we can observe that the attractor network is vital for the accuracy improvements. Our model achieves 5.8% accuracy improvement over the base model on the MIT-States dataset and 42.3% improvement on the UT-Zappos50K dataset, demonstrating the significance of attractor networks.

**Loss Compositions**    In this experiment, we evaluate the effects of different loss compositions, and the results are shown in Tab. 3. Since the attractor loss $L_{at}$ is indispensable for the attractor network and the fusion loss $L_{fus}$ is indispensable for the vision-language fusion mechanism, the sum of these two losses (al+fus) is taken as the basic loss. If only the basic loss is used, the accuracy is only 0.14% on the MIT-States dataset and 7.8% on the UT-Zappos50K dataset. When the discriminative loss (+dis) or conditional loss (+cd) is added to the basic loss, the accuracy slightly improves. When the decoding loss (+de) is added to the basic loss, our model achieves the striking accuracy improvements on both datasets, demonstrating that the encoder-decoder mechanism in our model is effective. We can also observe that the model tends to achieve higher accuracy when adding more losses, and the model obtains the highest accuracy when all losses are used, which further validates the effectiveness of the individual loss.

Table 4: Top-1 accuracies of three parameter sharing settings and our setting on the MIT-States dataset and UT-Zappos50k dataset. $f_a^{\mathcal{V}} = f_o^{\mathcal{V}}$ represents the parameter sharing in the visual encoding module, $f_a^{\mathcal{L}} = f_o^{\mathcal{L}}$ represents the parameter sharing in the linguistic encoding module, and $Att^{\mathcal{V}} = Att^{\mathcal{L}}$ represents the parameter sharing in the visual attractor network and linguistic attractor network.

| Parameter Sharing | MIT-States(%) | UT-Zappos(%) |
|:---:|:---:|:---:|
| $f_a^{\mathcal{V}} = f_o^{\mathcal{V}}$ | 37.8 | 60.1 |
| $f_a^{\mathcal{L}} = f_o^{\mathcal{L}}$ | 36.1 | 50.6 |
| $Att^{\mathcal{V}} = Att^{\mathcal{L}}$ | 2.4 | 1.6 |
| no sharing (our) | **38.2** | **73.7** |

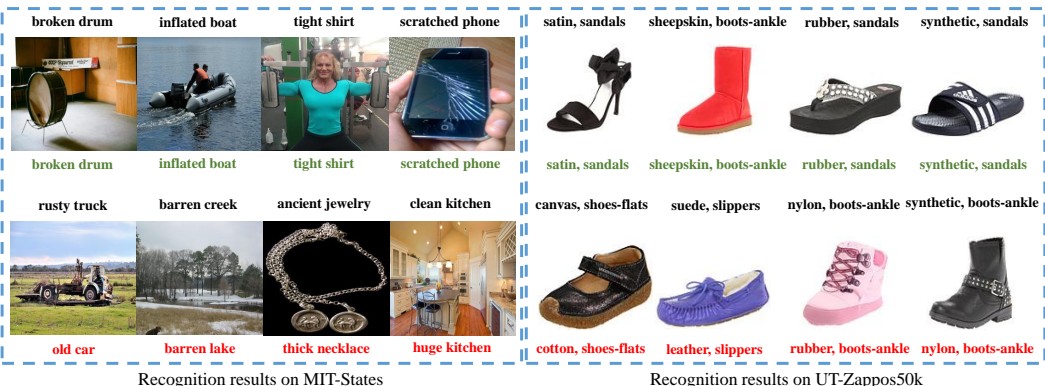

Figure 3: Samples of unseen attribute-object pair recognition on the MIT-States dataset (left) and UT-Zappos50k dataset (right).

**Parameter Sharing**   The attribute and object are fundamentally different entities, thus they should not be equally processed. Motivated by this idea, as illustrated in Fig. 2, we use different projection functions ($f_a^{\mathcal{V}}$ and $f_o^{\mathcal{V}}$) to obtain the attribute feature and the object feature in the visual encoding module, and we also use different projection functions ($f_a^{\mathcal{L}}$ and $f_o^{\mathcal{L}}$) to process the attribute feature and the object feature in the linguistic encoding module. In addition, the vision information differs from the language information in data structure, thus we use two attractor networks to separately process the visual feature and the linguistic feature. To validate our ideas, three parameter sharing experiments are conducted: 1) the attribute projection function $f_a^{\mathcal{V}}$ in the visual encoding module shares parameter with the object projection function $f_o^{\mathcal{V}}$ in the visual encoding module (i.e., let $f_a^{\mathcal{V}} = f_o^{\mathcal{V}}$), 2) the attribute projection function $f_a^{\mathcal{L}}$ in the linguistic encoding module shares parameter with the object projection function $f_o^{\mathcal{L}}$ in the linguistic encoding module (i.e., let $f_a^{\mathcal{L}} = f_o^{\mathcal{L}}$), and 3) the attractor network in the visual pathway $Att^{\mathcal{V}}$ shares the same parameter with the attractor network in the linguistic pathway $Att^{\mathcal{L}}$ (i.e., let $Att^{\mathcal{V}} = Att^{\mathcal{L}}$). The experiment results are shown in Tab. 4, and we can observe that our setting corresponds to the highest accuracies on two datasets, validating our ideas.

**Qualitative Result Analysis**   Fig. 3 shows some samples of recognitions on the MIT-States dataset (left) and UT-Zappos50k dataset (right). In the figure, the black texts on the top of images represent the ground truth, the green texts represent true recognitions, and the red texts represent false recognitions. We can observe that our model can correctly recognize some pairs with abstract attributes such as "tight shirt" and "satin sandal". However, as shown in the second row, some images are falsely recognized. One reason for false recognitions is that some attributes or objects present similar visual features. For example, the "rusty truck" is recognized as "old car" and "barren creek" is recognized as "barren lake". Actually, "rusty" and "old" exhibit the similar visual feature and "creek" shares the similar visual feature with "lake". Another reason is that one object type may be a subclass of another object type. For example, "ancient jewelry" is recognized as "thick necklace". Actually, the necklace is one kind of jewelry. In addition, one object having multiple attributes may also leads

to false recognitions. For example, "clean kitchen" is recognized as "huge kitchen". Actually, the kitchen indeed is huge and clean.

## 5  CONCLUSION

This paper studies a challenging and meaningful problem termed as unseen attribute-object pair recognition. To handle the problem, we propose a vision-language fusion generative model that involves attractor networks and the encoder-decoder mechanism. The proposed model presents impressive performance.

Our main conclusions are as follows: 1) Unseen attribute-object pair recognition is a complex problem that asks a model to learn the intrinsic feature representation and overcome the "domain shift" problem. 2) The attractor network presents the notable potentiality to learn an intrinsic feature representation and the encoder-decoder mechanism is significant for alleviating the "domain shift" problem. 3) During the testing, the voting inference method contributes to the accuracy improvements.

Unseen attribute-object pair recognition is a high-level vision problem, asking a model to be smart as humans. Therefore, we will further explore this problem using cognition-inspired methods in the future.

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
