# OpenReview forum: "BEYOND SUPERVISED LEARNING: RECOGNIZING UNSEEN ATTRIBUTE-OBJECT PAIRS WITH VISION-LANGUAGE FUSION AND ATTRACTOR NETWORKS"
_ICLR.cc/2020/Conference — Reject_

### Official Review · AnonReviewer1 · 2019-10-23
**Official Blind Review #1**

**Rating:** 1

**Review:**

The paper attempts to tackle unseen object-attribute recognition in still images. This follows a line of works that addresses such a problem using embedding (with operators), classification and generative approaches. The paper makes use of the visual attractor network, extracting different visual representations for objects and attributes. A collection of losses (some regularisation and some distance in embedding space) are proposed. I believe the paper (except A below) is technically correct.

This paper was particularly tricky to read. While convincing at parts, there are two worrying statements. One I currently believe (pending the authors' response) is a scientific flaw (A), the other causes the results to be potentially incomparable to published works (B). I detail these next.

(A) Scientific Flaw: Quoting from the paper, at inference (i.e. for a test image), "we propose a voting inference method. For an input image, its visual attractor feature AV is computed, ... By computing the L2 distance ... the pair that corresponds to the minimum distance is taken as the initial attribute-object pair recognition for the input image. For the images belonging to the same pair, we use the recognitions for all images to vote a pair label to be the final attribute-object pair recognition."
In my interpretation, the authors are taking all test images that are labeled using the same pair (e.g. 'sliced-apple' - assuming this is an unseen pair). They then use all these to make a final decision for all these images.
I believe my interpretation is correct because their statement "images belonging to the same pair" can only exist in testing for their target scenario (unseen object-attribute pair).
If my interpretation is correct, this is A MAJOR FLAW in the approach. How do the authors know that a collection of test images "belong to the same pair"?? This knowledge can only be acquired from labels of the test set and cannot be used as part of any algorithm.
I would very much like for my interpretation to be naive, but I could not find a different explanation for this statement in page 6. The authors note in the conclusion that this voting approach is part of their contribution.

(B) In reporting the results, the authors state that: "Using the grid-search method, we set parameters \alpha; ... [5 parameters]  in Eq.17 to be 1:0; 2:0; 6:0; 3:0; 2:0 for the MIT-States dataset and 1:0; 3:0; 6:0; 1:0; 2:0 for the UTZappos50K
dataset. Attractor parameters cv; cl; Tv; Tl are set as 0:1; 0:2; 20; 10 for the MIT-States dataset and 0:08; 0:16; 25; 15 for the UT-Zappos50K dataset."
This assumes that the authors are grid-searching all parameters, and evaluating the test set performance for each case, then reporting the maximum possible test performance per dataset. The authors do not reference a validation set they use to set these parameters instead. Importantly, the values are SIGNIFICANTLY different for each test set and there is no explanation of why the parameters vary largely. It is not clear how the performance on an new dataset would be,
Up to my knowledge, the other methods that they compare to in Table 1, have not reported a grid-search over the parameter space for maximum performance of the test set. This IMO makes the comparative evaluation, which the authors deem to be 'impressive' quite unfair. It is not possible to assess the method's performance using these results.

**Experience Assessment:**

I have published one or two papers in this area.

**Review Assessment: Checking Correctness Of Derivations And Theory:**

I carefully checked the derivations and theory.

**Review Assessment: Checking Correctness Of Experiments:**

I carefully checked the experiments.

**Review Assessment: Thoroughness In Paper Reading:**

I read the paper thoroughly.

---

### Official Review · AnonReviewer2 · 2019-10-25
**Official Blind Review #2**

**Rating:** 3

**Review:**

This paper studies the so-called “unseen attribute-object pair recognition”,
which asks a model to simultaneously recognize the attribute type and the
object type of a given image.

1) Despite the authours claimed this is a novel task, it has been thoroughly studied long time ago. For example, NEIL [1], and comparative attributes [2]. So it would be advisable to thoroughly review those related literature.

2) the whole framework is a two pathway encoder-decoder networks. The framework is well explained. It would be great if there are more words about the motivations why the network is designed in such a manner. In general, there is lack of discussion why the attractor networks are novel, and significant.

3) In term of network structure, the loss functions introduced are pretty straight-forward. Please claim novelty about loss functins, if there is anything special.

4) The experiments are pretty weak: Only on two small datasets. It would be great if there are more experiments on larger, or challenging datasets. Considering it’s almost ten pages; I would prefer more persuasive experiments to validate the novelties (but please better summarize the novelty again).



[1] NEIL: Extracting Visual Knowledge from Web Data. 2012

[2] Constrained Semi-Supervised Learning using Attributes and Comparative Attributes. ECCV 2012


**Experience Assessment:**

I have published in this field for several years.

**Review Assessment: Checking Correctness Of Derivations And Theory:**

I assessed the sensibility of the derivations and theory.

**Review Assessment: Checking Correctness Of Experiments:**

I carefully checked the experiments.

**Review Assessment: Thoroughness In Paper Reading:**

I read the paper at least twice and used my best judgement in assessing the paper.

---

### Official Review · AnonReviewer3 · 2019-11-04
**Official Blind Review #3**

**Rating:** 1

**Review:**

This paper tries to handle the unseen attribute-object pairs recognition, which asks a model to simultaneously recognize the attribute type and the object type of a given image while this attribute-object pair is not included in the training set. The claimed contribution includes: (1) they for the first time introduce the attractor networks to recognize unseen attribute-object pairs; (2) their method exhibits much better performance than conventional methods and state-of-the-art methods on two challenging public datasets.

I found this paper poorly written. First, the introduction of attractor networks lacks intuition, explanation, and experiment support. The only support of attractor network is Table2, which show marginal improvement on MIT-States and significant improvement on UT-Zappos. Even for the two picked dataset, the improvement is not consistent, this is not enough to show the effectiveness of attractor network. The authors could design some visualization/metrics to show what the attractor networks have learned, i.e., the visualization of the nodes. The authors should also provide more ablation studies on the attractor network, i.e., the number of the nodes, the time-tick Tv, etc. After all, the experiment results in this paper shows little evidence of how the attractor network works, and the insight of how it works if so.

Second, the ablation study and conclusion are confusing. Combining Table 1,2&3, we can see that the decoding loss is the magic. Therefore, I should say that the decoding loss is the main reason for the ''much better performance'' in this paper, instead of the attractor network. However, the baseline method GENERATE already has the reconstruction/decoder loss. Why is its number low?

I did not get the meaning of this sentence in Section 3.4. "For the images belonging to the same pair, we
use the recognitions for all images to vote a pair label to be the final attribute-object pair recognition."

**Experience Assessment:**

I have read many papers in this area.

**Review Assessment: Checking Correctness Of Derivations And Theory:**

N/A

**Review Assessment: Checking Correctness Of Experiments:**

I assessed the sensibility of the experiments.

**Review Assessment: Thoroughness In Paper Reading:**

I read the paper thoroughly.

---

### Decision · Program_Chairs · 2019-12-19

**Decision:**

Reject

**Comment:**

The paper focuses on attribute-object pairs image recognition, leveraging some novel "attractor network".

At this stage, all reviewers agree the paper needs a lot of improvements in the writing. There are also concerns regarding (i) novelty: the proposed approach being two encoder-decoder networks; (ii) lack of motivation for such architecture (iii) possible flow in the approach (are the authors using test labels?) and (iv) weak experiments.